# The bidirectional association between premenstrual disorders and perinatal depression: A nationwide register-based study from Sweden

Qian Yang[1]*, Emma Bränn[1], Elizabeth R. Bertone-Johnson[2,3], Arvid Sjölander[4], Fang Fang[1], Anna Sara Oberg[4], Unnur A. Valdimarsdóttir[1,5,6], Donghao Lu[1]*

1 Institute of Environmental Medicine, Karolinska Institutet, Stockholm, Sweden, 2 Department of Biostatistics and Epidemiology, School of Public Health and Health Sciences, University of Massachusetts Amherst, Amherst, Massachusetts, United States of America, 3 Department of Health Promotion and Policy, School of Public Health and Health Sciences, University of Massachusetts Amherst, Amherst, Massachusetts, United States of America, 4 Department of Medical Epidemiology and Biostatistics, Karolinska Institutet, Stockholm, Sweden, 5 Department of Epidemiology, Harvard TH Chan School of Public Health, Boston, Massachusetts, United States of America, 6 Center of Public Health Sciences, Faculty of Medicine, University of Iceland, Reykjavík, Iceland

* qian.yang.1@ki.se (QY); donghao.lu@ki.se (DL)

## Abstract

### Background

Premenstrual disorders (PMDs) and perinatal depression (PND) share symptomology and the timing of symptoms of both conditions coincide with natural hormonal fluctuations, which may indicate a shared etiology. Yet, there is a notable absence of prospective data on the potential bidirectional association between these conditions, which is crucial for guiding clinical management. Using the Swedish nationwide registers with prospectively collected data, we aimed to investigate the bidirectional association between PMDs and PND.

### Methods and findings

With 1,803,309 singleton pregnancies of 1,041,419 women recorded in the Swedish Medical Birth Register during 2001 to 2018, we conducted a nested case-control study to examine the risk of PND following PMDs, which is equivalent to a cohort study, and transitioned that design into a matched cohort study with onward follow-up to simulate a prospective study design and examine the risk of PMDs after PND (within the same study population). Incident PND and PMDs were identified through clinical diagnoses or prescribed medications. We randomly selected 10 pregnant women without PND, individually matched to each PND case on maternal age and calendar year using incidence density sampling (N: 84,949: 849,482). We (1) calculated odds ratio (OR) and 95% confidence intervals (CIs) of PMDs using conditional logistic regression in the nested case-control study. Demographic factors (country of birth, educational level, region of residency, and cohabitation status) were adjusted for. We (2) calculated the hazard ratio (HR) and 95% CIs of PMDs subsequent to PND using stratified Cox regression in the matched cohort study. Smoking, BMI, parity, and

**Data Availability Statement:** Data are from the Swedish national healthcare registers. Data cannot be put into a public data repository according to Swedish law but are available by applying through Statistics Sweden or the Swedish National Board of Health and Welfare. Detailed information on data

application can be found in the following links: https://www.scb.se/vara-tjanster/bestalla-mikrodata/ and https://bestalladata.socialstyrelsen.se/.

**Funding:** The work was supported by the Chinese Scholarship Council (No. 201700260289 to QY), the Swedish Research Council for Health, Working Life and Welfare (FORTE) (No. 2020-00971 and 2023-00399 to DL), the Swedish Research Council (Vetenskapsrådet) (No. 2020-01003 to DL), Karolinska Institutet Strategic Research Area in Epidemiology and Biostatistics (grant to DL), Karolinska Institutet SFOepi Junior Scholar Grant (to DL) and the Icelandic Research Fund (No. 218274-051 to UAV). The funders had no role in study design, data collection and analysis, preparation of the manuscript or decision to publish.

**Competing interests:** The authors have declared that no competing interests exist.

**Abbreviations:** ACOG, American College of Obstetricians and Gynecologists; CI, confidence interval; HR, hazard ratio; IR, incidence rate; MBR, Medical Birth Register; MGR, Multi-Generation Registers; OR, odds ratio; PMD, premenstrual disorder; PMDD, premenstrual dysphoric disorder; PMS, premenstrual syndrome; PND, perinatal depression.

history of psychiatric disorders were further controlled for, in addition to demographic factors. Pregnancies from full sisters of PND cases were identified for sibling comparison, which contrasts the risk within each set of full sisters discordant on PND. In the nested case-control study, we identified 2,488 PMDs (2.9%) before pregnancy among women with PND and 5,199 (0.6%) among controls. PMDs were associated with a higher risk of subsequent PND (OR 4.76, 95% CI [4.52,5.01]; $p < 0.001$). In the matched cohort with a mean follow-up of 7.40 years, we identified 4,227 newly diagnosed PMDs among women with PND (incidence rate (IR) 7.6/1,000 person-years) and 21,326 among controls (IR 3.8). Compared to their matched controls, women with PND were at higher risk of subsequent PMDs (HR 1.81, 95% CI [1.74,1.88]; $p < 0.001$). The bidirectional association was noted for both prenatal and postnatal depression and was stronger among women without history of psychiatric disorders ($p$ for interaction $< 0.001$). Sibling comparison showed somewhat attenuated, yet statistically significant, bidirectional associations. The main limitation of this study was that our findings, based on clinical diagnoses recorded in registers, may not generalize well to women with mild PMDs or PND.

## Conclusions

In this study, we observed a bidirectional association between PMDs and PND. These findings suggest that a history of PMDs can inform PND susceptibility and vice versa and lend support to the shared etiology between both disorders.

## Author summary

### Why was this study done?

- Perinatal depression (PND) and premenstrual disorders (PMDs) share symptomology (e.g., feeling depressed), and the timing of symptom onset of both conditions coincides with natural hormonal fluctuations.

- Prospective data are lacking to study the potential bidirectional association between these conditions, which can guide clinical management.

### What did the researchers do and find?

- We conducted a nested case-control study and transitioned that design into a matched cohort study with onward follow-up to simulate a prospective study design.

- Among approximately 1.8 million singleton pregnancies in Sweden during 2001 to 2018, we identified 84,949 women with PND and 849,482 unaffected women, individually matched on age and calendar year. Pregnancies from full sisters of women with PND were also identified for sibling comparison.

- Among women with PND, 2.9% had PMDs before pregnancy, in contrast to 0.6% among matched unaffected women. PMDs were associated with a nearly 5 times higher risk of subsequent PND. In the matched cohort with a mean follow-up of 6.90 years,

women with PND had almost 2 times higher risk of subsequent PMDs, compared to matched unaffected women.

- The bidirectional association between PMDs and PND was noted for both prenatal and postnatal depression, regardless of history of psychiatric disorders, and also in sibling comparison.

## What do these findings mean?

- These findings suggest that a history of PMDs can inform PND susceptibility and vice versa.

- The main limitation of this study was that our findings, based on clinical diagnoses or prescribed medications, may not generalize well to women with mild PMDs or PND.

## Introduction

Premenopausal women experience natural hormonal fluctuations associated with various life events, such as puberty, menstrual cycle, pregnancy, and menopause. Some women are more likely to develop or manifest mood symptoms during these events. For instance, perinatal depression (PND) is characterized by depressive symptoms occurring during pregnancy and up to 12 months after delivery and affects 11% of mothers globally [1]. PND has been positively associated with maternal suicidal behavior and has a negative influence on mother–infant bonding [2]. Similarly, premenstrual disorders (PMDs) are characterized by somatic and/or psychological symptoms that recur in luteal phase. PMDs cause significant functional impairment [3–5]. PMDs affect 20% to 30% of women of reproductive age [4], and about 5% to 8% of women suffer from severe symptomology [3]. PMDs are associated with increased risks of suicidal behavior and accidents [6].

PND and PMDs share symptomology (e.g., feeling depressed) and the timing of symptom onset of both conditions coincides with natural hormonal fluctuations [7,8]. Therefore, it has been postulated that these disorders may have common etiology and shared risk factors [9]. This hypothesis is supported by 2 recent systematic reviews showing that women with PND were more likely to have a history of PMDs [10,11]. However, existing studies relied on retrospectively collected on data premenstrual symptoms during or after pregnancy, which might be prone to recall bias and thereby biased results [12,13]. Moreover, the community- or clinic-based sampling in previous studies may have introduced significant selection bias [14]. Premenstrual symptoms can worsen after pregnancy due to an escalated abnormal response to hormonal changes in relation to pregnancy [15]. It is thus plausible that women with PND are at risk for subsequent PMDs. However, few studies with a relatively small sample size have examined this hypothesis by comparing proportions without adjustment for confounders and generated inconsistent results [16–18].

Taken together, without prospective evidence, it remains unclear whether women with PMDs have an increased risk of developing PND when becoming pregnant or after giving birth and vice versa. Using the Swedish nationwide registers with prospectively collected data, we aimed to investigate the bidirectional association between PMDs and PND. To study the bidirectional association in the same population effectively, we conducted a nested case-

control study, a design inherently equivalent to a cohort study [19]. To examine the risk of PND following PMDs in a manner that simulates a prospective approach, we then transitioned this design into a matched cohort study to assess the risk of PMDs after PND. We further employed sibling comparisons to account for shared genetic and familial environmental risk factors for both disorders.

## Methods

### Data sources

Based on the Medical Birth Register (MBR), we identified 1,803,309 singleton pregnancies from 1,041,419 women during 2001 to 2018. The MBR covers virtually all births in Sweden since 1973, with rich information prospectively collected from prenatal, delivery, and neonatal care [20,21]. Multiple births ($n$ = 51,824), pregnancies after PND diagnosis ($n$ = 34,790), pregnancies from women who emigrated before 2001 ($n$ = 952), or pregnancies before age 15 or after age 52 ($n$ = 383) were excluded. The Patient Register, Prescribed Drug Register, Migration Register, Causes of Death Register, and Multi-Generation Register (MGR) were cross-linked using the unique personal identification number. The Patient Register collects information on all inpatient admissions for psychiatric care since 1973 and for somatic diseases since 1987 and >80% outpatient visits since 2001 [22]. The Prescribed Drug Register contains information on medications redeemed from all pharmacies in Sweden since July 2005 [23]. MGR contains information on familial links for individuals born from 1932 onward [24].

### Ascertainment of PND

In line with previous studies [25], we identified PND from the date obtained by subtracting gestational age from the delivery date till 1 year postpartum, using the Swedish version of International Statistical Classification of Diseases and Related Health Problems (ICD)-10 codes (F32, F33, and F53.0) recorded in the MBR and Patient Register. According to the Swedish National Board of Health and Welfare, F32 and F33 are ICD-10 codes used to identify depression in Swedish healthcare registers according to the Swedish National Board of Health and Welfare [26] and have been reported to have high validity in the Swedish population (κ of 0·32; 88% full agreement with gold standard) [27]. F53.0 identifies perinatal depression not captured elsewhere [26]. Gestational age was, whenever possible, estimated according to ultrasound, which has been offered to all pregnant women in Sweden since 1990 and is performed for 95% of all pregnancies [28]. Nearly half of mental health problems are managed in primary care in Sweden [29]. Therefore, any prescription of antidepressants (ATC code N06A) was also considered as a proxy for PND. Antidepressants are commonly prescribed for PMDs as first-line treatment [30]. Prescriptions of antidepressants with an indication for PMDs, as described elsewhere [6], were not considered. The date of PND diagnosis was defined as the date of receiving a clinical diagnosis or filling a prescription of antidepressants, whichever came first. Since the MBR does not record the date of diagnosis and/or drug use, the median date of pregnancy was assigned as diagnosis date for those identified through MBR. Perinatal depression was then subcategorized into prenatal and postnatal depression using delivery date as cutoff point.

### Ascertainment of PMDs

Clinical diagnoses of PMDs were retrieved from the Patient Register (ICD-10 code N943). PMDs include premenstrual syndrome (PMS) and premenstrual dysphoric disorder (PMDD). In practice, PMS is often diagnosed based on criteria similar to the American College of

Obstetricians and Gynecologists (ACOG) criteria [31], and PMDD is diagnosed in accordance with diagnostic criteria described in DSM-5 [32]. According to the Swedish guidelines, prospective daily symptom ratings for at least 2 consecutive menstrual cycles are required [33]. To capture diagnoses made in primary care, we identified all prescriptions of antidepressants (ATC codes N06AA, N06AB, and N06AX) and contraceptives (G02B and G03A) with a specified clinical indication of PMDs from the Prescribed Drug Register. Indications of PMDs were specified by the prescribers as free-text and identified with key word recognition, as described previously [6].

## Study design

We identified 84,949 incident cases of PND, including 47,424 cases of prenatal depression and 37,525 cases of postnatal depression. Using incidence density sampling, 10 controls that were free from PND at the time when the matched case was diagnosed were randomly selected for each case. Controls were matched on gestational age for prenatal depression cases and matched on postnatal day for postnatal depression cases, together with maternal age (*n* = 849,482). To effectively examine the bidirectional association between PMDs and PND within the same study population, we conducted a nested case-control study, a design inherently equivalent to a cohort study [19], identifying all pregnancies, corresponding PNDs and previous history of any indications for PMDs to examine the risk of PND following PMDs in a manner that simulates a prospective approach. We then transitioned this design into a matched cohort with onward follow-up of index PNDs and control pregnancies, enabling us to efficiently assess the risk of PMDs after PND within the same study population. The matching date was used as the index date. Women who had PMDs before their index date were excluded (*n* = 7,687) and, as they were not at risk for incident PMD diagnosis. All women were then followed from 6 months after delivery (by when over 90% of Swedish women had stopped complete breastfeeding) [34] or the index date, whichever came later, until age 52, emigration, PMD diagnosis, or end of follow-up, whichever came first. During the follow-up, we observed 500 deaths in PND group and 1,559 deaths in matched controls.

The study design is illustrated in S1 Fig. The study was approved by the Regional Ethics Review Board in Stockholm (No. 2018-1515/31). Written informed consent is waived for register-based studies by Swedish law. This study is reported as per the Strengthening the Reporting of Observational Studies in Epidemiology (STROBE) guideline (S1 Checklist).

## Covariates

Information on the following covariates were retrieved: demographics (maternal age, country of birth, cohabitation status, region of residence, and educational level), smoking 3 months before the pregnancy, BMI in early pregnancy, parity, history of psychiatric disorders before pregnancy, and pregnancy complications and adverse outcomes including hypertensive and diabetic diseases, preterm birth (gestational week <37 weeks), stillbirth, low birth weight (birth weight<2,500 grams), congenital malformations of the offspring, and neonatal death of the offspring (death within 28 days of birth). Data origin and rational of the choice of covariates are described in S1 Methods. ICD codes are summarized in S1 Table, and categorization for covariates are presented in Table 1. Missing values in covariates were coded as unknown for adjustment.

## Statistical analysis

We firstly compared the distributions of characteristics between PND cases and their matched controls.

**Table 1. Characteristics of women with and without perinatal depression (PND).**

| | Control | PND | p |
|---|---|---|---|
| Number | 849,482 | 84,949 | |
| **Matching variables** | | | |
| Maternal age at pregnancy | | | 0.755 |
| 15–24 | 34,353 (4.0) | 3,460 (4.1) | |
| 25–34 | 423,729 (49.9) | 42,315 (49.8) | |
| 35–44 | 369,579 (43.5) | 36,944 (43.5) | |
| 45–52 | 21,821 (2.6) | 2,230 (2.6) | |
| Calendar year of pregnancy | | | 0.914 |
| 2000–2004 | 76,350 (8.99) | 7,620 (8.97) | |
| 2005–2009 | 250,584 (29.5) | 25,123 (29.57) | |
| 2010–2014 | 306,657 (36.1) | 30,702 (36.14) | |
| 2015–2018 | 215,891 (25.41) | 21,504 (25.31) | |
| **Demographics** | | | |
| Cohabitation status | | | <0.001 |
| Yes | 754,536 (88.8) | 71,064 (83.7) | |
| No | 15,966 (1.9) | 3,416 (4.0) | |
| Unknown | 78,980 (9.3) | 10,469 (12.3) | |
| Country of birth | | | <0.001 |
| Sweden | 637,192 (75.0) | 71,867 (84.6) | |
| Others | 212,208 (25.0) | 13,077 (15.4) | |
| Unknown | 82 (0.0) | 5 (0.0) | |
| Highest educational level | | | <0.001 |
| Primary | 250,882 (29.5) | 30,928 (36.4) | |
| High school | 311,924 (36.7) | 32,243 (38.0) | |
| College and beyond | 250,144 (29.4) | 19,257 (22.7) | |
| Unknown | 36,532 (4.3) | 2,521 (3.0) | |
| Region of residence | | | <0.001 |
| South | 197,509 (23.3) | 19,254 (22.7) | |
| Middle | 498,808 (58.7) | 49,926 (58.8) | |
| North | 146,658 (17.3) | 15,600 (18.4) | |
| **Pregnancy characteristics** | | | |
| Smoking before pregnancy | | | <0.001 |
| No smoking | 680,606 (80.1) | 57,696 (67.9) | |
| 1–9 cigarette per day | 62,631 (7.4) | 9,197 (10.8) | |
| ≥10 cigarette per day | 61,190 (7.2) | 13,531 (15.9) | |
| Unknown | 45,055 (5.3) | 4,525 (5.3) | |
| BMI in early pregnancy (kg/m$^2$) | | | <0.001 |
| <18.5 | 19,934 (2.3) | 2,031 (2.4) | |
| 18.5 to 24.9 | 468,061 (55.1) | 42,238 (49.7) | |
| 25 to 29.9 | 197,301 (23.2) | 20,719 (24.4) | |
| ≥30 | 97,711 (11.5) | 13,100 (15.4) | |
| Unknown | 66,475 (7.8) | 6,861 (8.1) | |
| Diabetic diseases | | | <0.001 |
| No | 832,688 (98.0) | 82,408 (97.0) | |
| Diabetes | 7,615 (0.9) | 1,392 (1.6) | |
| Gestational diabetes | 9,179 (1.1) | 1,149 (1.4) | |
| Hypertensive diseases | | | <0.001 |

*(Continued)*

**Table 1.** (Continued)

| | Control | PND | *p* |
|---|---|---|---|
| No | 822,521 (96.8) | 81,250 (95.6) | |
| Essential hypertension | 3,339 (0.4) | 436 (0.5) | |
| Preeclampsia | 23,622 (2.8) | 3,263 (3.8) | |
| History of psychiatric disorders before the pregnancy | | | <0.001 |
| No | 758,311 (89.3) | 39,232 (46.2) | |
| Depression | 30,989 (3.7) | 22,939 (27.0) | |
| Others | 60,182 (7.1) | 22,778 (26.8) | |
| Parity | | | <0.001 |
| 1 | 384,770 (45.3) | 44,792 (52.7) | |
| 2 | 306,442 (36.1) | 24,349 (28.7) | |
| 3 | 110,336 (13.0) | 10,845 (12.8) | |
| ≥4 | 47,934 (5.6) | 4,963 (5.8) | |
| Pregnancy outcomes | | | |
| Mode of delivery | | | <0.001 |
| Cesarean section | 138,242 (16.3) | 18,836 (22.2) | |
| Assisted vaginal delivery | 57,968 (6.8) | 6,222 (7.3) | |
| Nonassisted vaginal delivery | 653,272 (76.9) | 59,891 (70.5) | |
| Preterm birth | | | <0.001 |
| No | 809,231 (95.3) | 78,896 (92.9) | |
| Yes | 40,251 (4.7) | 6,053 (7.1) | |
| Low birth weight[1] | | | <0.001 |
| No | 820,694 (96.6) | 80,737 (95.0) | |
| Yes | 27,596 (3.2) | 4,042 (4.8) | |
| Offspring death | | | <0.001 |
| No | 845,480 (99.6) | 84,093 (91.1) | |
| Stillbirth | 2,817 (0.3) | 614 (0.7) | |
| Neonatal death[2] | 1,185 (0.1) | 242 (0.3) | |
| Congenital malformations | | | <0.001 |
| No | 780,100 (91.8) | 77,236 (90.9) | |
| Yes | 69,382 (8.2) | 7,713 (9.1) | |

[1]Birth weight <2,500 grams.

[2]Death within 28 days of birth.

*p*-Values were obtained from chi-squared test.

**PMDs and subsequent risk of PND.** In the nested case-control study, we calculated the percentage of PMDs before pregnancy for PND cases and controls separately and estimated odds ratios (ORs) and 95% confidence intervals (CIs) using conditional logistic regression. The nested case–control analysis, by design, is equivalent to the analysis using full cohort (i.e., assessing the incidence of PND subsequent to PMDs). Using incidence density sampling for individual matching, the estimate OR can be interpreted as hazard ratio (HR) of PND, comparing women with and without PMDs before pregnancy [19]. This analysis was repeated separately for prenatal and postnatal depression, and PND diagnosed in different time windows [first trimester (within 13 gestational weeks) or second to third trimester for prenatal depression, and 0 to 3, 4 to 6, or 7 to 12 months after delivery for postnatal depression].

Both PMDs and PND are correlated with depression and other psychiatric disorders [35,36]. An interaction term between history of psychiatric disorders and PND was added to

explore potential effect modification. Because parity is associated with many pregnancy complications and outcomes [37], we performed separate analyses for primi- and multiparous women. To explore effect modification by maternal age and calendar year, we also conducted stratified analysis by maternal age (categorized into 15 to 30 and 31 to 52) and calendar year at pregnancy.

**PND and subsequent risk of PMDs.**   In the matched cohort study, we calculated the incidence rate (IR) of PMDs among PND cases and matched controls and HRs and 95% CIs of PMDs using Cox regression (attained age as the underlying time scale and matching sets as strata). The proportional hazard assumption was deemed reasonable by inspecting the Schoenfeld residuals. Consistent with the nested case-control analyses, we performed analyses by PND subtype (pre- and postnatal depression and PND diagnosed in different time windows) and other stratification analyses.

**Adjustment.**   Model 1 accounted for the matching variables and Model 2 accounted for other demographic factors (country of birth, educational level, region of residency, and cohabitation status). In Model 3, smoking, BMI, parity, and history of psychiatric disorders were further controlled for. We considered Model 3 as the primary for the matched cohort study and Model 2 for the nested case-control study since these covariates were instead possible mediators in this scenario.

**Sibling analysis.**   PMDs and PND may have shared risk factors, such as poor support from family members and genetic factors [10,11], which would confound the studied associations. Sibling comparison contrasts the risk within each set of full sisters discordant on PND and inherently controls for unmeasured confounders shared between full sisters [38]. Briefly, all pregnancies from PND cases and their full sisters were identified through MBR linked to MGR. In total, 56,941 pregnancies from 40,665 full sisters (18,869 PND cases) were included. We examined the bidirectional association using conditional logistic regression and stratified Cox regression.

**Additional analyses.**   We limited the analysis to (1) PMDs with at least 2 specialists' PMD diagnoses ≥28 days apart to test the validity of PMD diagnosis; (2) PND identified through clinical diagnosis alone to reduce misclassification by using dispensation of antidepressants; and (3) women without severe pregnancy complications or adverse delivery outcomes since they might confound or mediate the studied associations through chronic stress associated with such events [39]. Due to lack of individual-level data on return of postpartum menstruation, we performed a sensitivity analysis of cohort follow-up starting from 2, 3, or 12 months postpartum.

Data were prepared in SAS statistical software version 9.4 (SAS Institute, Cary, NC) and analyzed in Stata 15.1 (STATA, College Station, TX). The statistical significance was set at the nominal two-sided 5% level.

## Results

### Characteristics

The median age was 30.71 at PND diagnosis. Compared to the controls, women with PND were less educated, were more likely to be born in Sweden, live alone and in South Sweden, had a higher BMI, were more likely to smoke, and have been diagnosed with psychiatric disorder before the pregnancy (all *p*-values < 0.001; Table 1). They were also more likely to be primiparous, deliver through cesarean section, and experience pregnancy complications and adverse pregnancy and birth outcomes (all *p*-values < 0.001; Table 1).

**Table 2. Association of premenstrual disorders (PMDs) with subsequent risk of perinatal depression (PND): A nested case-control study.**

| | Women without PND | Women with PND | Model 1[1] | | Model 2[2] | |
|---|---|---|---|---|---|---|
| | N (%) of PMDs | N (%) of PMDs | OR (95% CI) | p | OR (95% CI) | p |
| Overall | 5,199 (0.6) | 2,488 (2.9) | 4.98 (4.74,5.23) | <0.001 | 4.76 (4.52,5.01) | <0.001 |
| *By time of diagnosis* | | | | | | |
| Prenatal depression | 3,052 (0.6) | 1,408 (3.0) | 4.80 (4.50,5.12) | <0.001 | 4.58 (4.28,4.90) | <0.001 |
| *By time since pregnancy* | | | | | | |
| First trimester | 2,089 (0.8) | 870 (3.1) | 4.32 (3.98,4.68) | <0.001 | 3.90 (3.59,4.24) | <0.001 |
| Second-third trimester | 963 (0.5) | 538 (2.7) | 5.86 (5.26,6.53) | <0.001 | 5.54 (4.96,6.19) | <0.001 |
| Postnatal depression | 2,147 (0.6) | 1,080 (2.9) | 5.23 (4.85,5.63) | <0.001 | 5.03 (4.65,5.45) | <0.001 |
| *By time since delivery* | | | | | | |
| ≤6 months | 1,044 (0.6) | 437 (2.4) | 4.31 (3.85,4.83) | <0.001 | 4.10 (3.66,4.61) | <0.001 |
| 7–12 months | 1,103 (0.6) | 643 (3.3) | 6.10 (5.52,6.74) | <0.001 | 5.80 (5.24,6.42) | <0.001 |

CIs, confidence intervals; N, number; OR, odds ratio; PMDs, premenstrual disorders; PND, perinatal depression.

ORs and p-values were obtained from conditional logistic regression.

[1]Model 1 was adjusted for the matching variable (i.e., maternal age and calendar year).

[2]Model 2 was additionally adjusted for country of birth (Sweden or not), educational level (primary, high school, college and beyond), region of residence (south, middle, or north of Sweden), and cohabitation status (yes or no) at matching.

## PMDs and subsequent risk of PND

We identified 7,687 women with PMDs before pregnancy (2,488 among women with PND) in the nested case-control study. PMDs were associated with a higher risk of subsequent PND (OR 4.98, 95% CI [4.74,5.23]; $p < 0.001$). Additional adjustment of demographic factors including country of birth, educational level, region of residence, and cohabitation status slightly attenuated the observed association (OR 4.76, 95% CI [4.52,5.01]; $p < 0.001$) (Table 2). The association was observed for both prenatal (OR 4.58, 95% CI [4.28,4.90]; $p < 0.001$) and postnatal (OR 5.03, 95% CI [4.65,5.45]; $p < 0.001$) depression. Moreover, the association remained robust across different pre-/postnatal phases; the OR was lower for prenatal depression diagnosed during first trimester than for those diagnosed later during pregnancy and lower for postnatal depression within 6 months after delivery than those diagnosed during 7 to 12 months (Table 2). Attenuated but statistically significant results were observed after adjusting for potential mediators including history of psychiatric disorders (S2 Table).

In the stratified analysis, an association between PMDs and subsequent PND was observed regardless of previous history of psychiatric disorders and was stronger among women without such history ($p$ for interaction $< 0.001$) (Table 3). The association was stronger among multiparous than primiparous women and was comparable across maternal age and calendar year groups (S3 Table).

## PND and subsequent risk of PMDs

During a mean follow-up of 6.90 years (standardized deviation 4.31, range 0.003 to 17.50) from 6 months postpartum onwards, we identified 25,553 newly diagnosed cases of PMDs (4,227 among women with PND) in the matched cohort study. Compared to their matched controls, women with PND were at higher risk of PMDs (HR 2.00, 95% CI [1.93,2.07]; $p < 0.001$). Adjustment for pregnancy characteristics, history of psychiatric disorders, together with demographic factors slightly attenuated the association (HR 1.81, 95% CI [1.74,1.88]; $p < 0.001$) (Table 4). Estimated HRs were 1.67, 95% CI [1.58,1.76]; $p < 0.001$ for prenatal and

**Table 3. Association of premenstrual disorders (PMDs) with subsequent risk of perinatal depression (PND), stratified by history of psychiatric disorders before pregnancy: A nested case control study.**

| | Women without PND | Women with PND | Model 1[1] | | Model 2[2] | |
|---|---|---|---|---|---|---|
| | N (%) of PMDs | N (%) of PMDs | OR (95% CI) | p | OR (95% CI) | p |
| *By history of psychiatric disorder* | | | | | | |
| No | 3,940 (0.5) | 1,481 (2.9) | 6.03 (5.65,6.44) | <0.001 | 5.73 (5.35,6.14) | <0.001 |
| Depression | 296 (2.1) | 329 (3.2) | 1.61 (1.34,1.93) | <0.001 | 1.67 (1.38,2.03) | <0.001 |
| Other disorders | 963 (1.6) | 678 (3.0) | 1.95 (1.74,2.18) | <0.001 | 1.95 (1.73,2.20) | <0.001 |
| *p* for interaction | | | | <0.001 | | <0.001 |

CIs, confidence intervals; N, number; OR, odds ratio; PMDs, premenstrual disorders; PND, perinatal depression.

ORs and *p*-values were obtained from logistic regression.

[1]Model 1 was adjusted for the matching variable (i.e., maternal age and calendar year).

[2]Model 2 was additionally adjusted for country of birth (Sweden or not), educational level (primary, high school, college and beyond), region of residence (south, middle, or north of Sweden), and cohabitation status (yes or no) at matching.

1.98, 95% CI [1.87,2.09]; *p* < 0.001 for postnatal depression. Similar associations were found across time windows during pregnancy and after delivery (Table 4).

The association between PND and subsequent PMDs was stronger among women without a history of psychiatric disorder in the stratified analysis (*p* for interaction < 0.001) (Table 5). The association did not differ by parity and calendar year and was somewhat greater among women aged 31 to 52 years at pregnancy (S4 Table).

## Additional analyses

Largely comparable bidirectional associations between PMDs and PND were observed in the sibling comparison, although results were somewhat attenuated compared to the population comparison (Table 6).

**Table 4. Association of perinatal depression (PND) with subsequent risk of premenstrual disorders (PMDs): A matched cohort study.**

| | Women without PND | Women with PND | Model 1[1] | | Model 2[2] | | Model 3[3] | |
|---|---|---|---|---|---|---|---|---|
| | N (IR) of PMDs | N (IR) of PMDs | HR (95% CI) | p | HR (95% CI) | p | HR (95% CI) | p |
| Overall | 21,326 (3.8) | 4,227 (7.6) | 2.00 (1.93,2.07) | <0.001 | 1.97 (1.90,2.04) | <0.001 | 1.81 (1.74,1.88) | <0.001 |
| *By time of diagnosis* | | | | | | | | |
| Prenatal depression | 11,465 (3.8) | 2,249 (7.2) | 1.90 (1.82,1.99) | <0.001 | 1.87 (1.78,1.96) | <0.001 | 1.67 (1.58,1.76) | <0.001 |
| *By time since pregnancy* | | | | | | | | |
| First trimester | 6,073 (4.2) | 1,151 (7.6) | 1.81 (1.70,1.93) | <0.001 | 1.73 (1.62,1.85) | <0.001 | 1.52 (1.40,1.65) | <0.001 |
| Second-third trimester | 5,392 (3.4) | 1,098 (6.8) | 2.01 (1.88,2.15) | <0.001 | 1.98 (1.85,2.12) | <0.001 | 1.81 (1.67,1.95) | <0.001 |
| Postnatal depression | 9,382 (3.9) | 1,978 (8.2) | 2.12 (2.02,2.23) | <0.001 | 2.09 (1.98,2.20) | <0.001 | 1.98 (1.87,2.09) | <0.001 |
| *By time since delivery* | | | | | | | | |
| ≤6 months | 4,386 (3.9) | 873 (7.7) | 1.98 (1.84,2.14) | <0.001 | 1.94 (1.80,2.09) | <0.001 | 1.81 (1.67,1.97) | <0.001 |
| 7–12 months | 4,996 (3.9) | 1,105 (8.7) | 2.24 (2.09,2.39) | <0.001 | 2.21 (2.06,2.36) | <0.001 | 2.13 (1.98,2.30) | <0.001 |

CIs, confidence intervals; HR, hazard ratio; IR, incidence rate, per 1,000 person-years; N, number; PMDs, premenstrual disorders; PND, perinatal depression.

HRs and *p*-values were obtained from conditional Cox regression.

[1]Model 1 was adjusted for the matching variable (i.e., maternal age and calendar year).

[2]Model 2 was additionally adjusted for country of birth (Sweden or not), educational level (primary, high school, college and beyond), region of residence (south, middle, or north of Sweden), and cohabitation status (yes or no) at matching.

[3]Model 3 was additionally adjusted for parity (1, and ≥2), BMI during early pregnancy (categorized into <18.5, 18.5 to 24.9, 25 to 29.9, and ≥30 kg/m$^2$), and smoking before pregnancy (no smoking, 1–9, and ≥10 cigarettes per day) and history of psychiatric disorders before pregnancy (yes or no).

**Table 5. Association of perinatal depression (PND) with subsequent risk premenstrual disorders (PMDs) stratified by history of psychiatric disorders: A matched cohort study.**

| | Women without PND | Women with PND | Model 1[1] | | Model 3[2] | |
|---|---|---|---|---|---|---|
| | N (IR) of PMDs | N (IR) of PMDs | HR (95% CI) | *p* | HR (95% CI) | *p* |
| *By history of psychiatric disorders* | | | | | | |
| No | 19,044 (3.7) | 2,709 (7.6) | 2.10 (2.01,2.19) | <0.001 | 2.08 (1.99,2.18) | <0.001 |
| Depression | 497 (8.4) | 464 (7.8) | 0.97 (0.84,1.12) | 0.684 | 0.94 (0.80,1.09) | 0.394 |
| Other disorders | 1,785 (5.9) | 1,054 (7.7) | 1.32 (1.21,1.44) | <0.001 | 1.34 (1.23,1.47) | <0.001 |
| *p* for interaction | | | | <0.001 | | <0.001 |

CIs, confidence intervals; HR, hazard ratio; IR, incidence rate, per 1,000 person-years; N, number; PMDs, premenstrual disorders; PND, perinatal depression.

HRs and *p*-values were obtained from conditional Cox regression.

[1]Model 1 was adjusted for the matching variable (i.e., maternal age and calendar year).

[2]Model 3 was additionally adjusted for country of birth (Sweden or not), educational level (primary, high school, college and beyond), region of residence (south, middle, or north of Sweden), and cohabitation status (yes or no) at matching, parity (1, and ≥2), BMI during early pregnancy (categorized into <18.5, 18.5 to 24.9, 25 to 29.9, and ≥30 kg/m$^2$), and smoking before pregnancy (no smoking, 1–9, and ≥10 cigarettes per day) and history of psychiatric disorders before pregnancy (yes or no).

In both study designs, robust bidirectional associations were observed when restricting to (1) women without pregnancy complications or adverse outcomes; (2) PMDs with 2 specialists' diagnoses ≥28 days apart; and (3) PND ascertained through clinical diagnosis (S5 Table). Moreover, similar results were observed with different start of follow-up in the cohort study (S6 Table).

## Discussion

In the present study based on data from Swedish national registers, we found a bidirectional link between premenstrual disorders and perinatal depression, which was pronounced for prenatal and postnatal depression and slightly stronger for postnatal depression. The bidirectional association was verified among women without a history of psychiatric disorders. These findings were corroborated, despite the attenuation, in the sibling comparison, which controls for familial environmental and genetic factors.

Previous studies have used data on history of PMDs retrospectively collected during or after pregnancy, which might be vulnerable to recall bias and systematic error [12,13]. Taking advantage of prospectively collected data in Swedish healthcare registers, the study is the first, to our knowledge, to demonstrate a positive association between PMDs and subsequent risk of PND. Moreover, previous studies exclusively focused on postnatal depression, mostly ascertained within 8 weeks postpartum [10,11]. With retrospectively collected data, one systematic review and one meta-analysis reported that women with postnatal depression were more likely to endorse a history of PMDs [10,11], which is in line with our results. The meta-analysis reported an OR of 2.20, 95% CI [1.81,2.68] for PMDs among women with postnatal depression (mostly assessed within 6 months after delivery) [10]. Correspondingly, our data show that women with PMDs have more than 3 times higher risk of PND within 6 months postpartum.

To our knowledge, no study has examined the association between prenatal depression and PMDs. Our findings on prenatal depression therefore extend knowledge to another subtype of PND that occurs during pregnancy. Although the etiology for prenatal depression is complex [40], our finding may suggest a subgroup of prenatal depression may be related to hormone changes as well. However, future studies are needed to confirm our results and to understand the biologic link between PMDs and prenatal depression. Interestingly, we found a stronger association between PMDs and PND among multiparous than primiparous women. It is

**Table 6. Bidirectional association between premenstrual disorders (PMDs) and perinatal depression (PND) in sibling comparison.**

| PMDs and subsequent risk of PND | | | | | | |
|---|---|---|---|---|---|---|
| | Women without PND | Women with PND | Model 1[1] | | Model 2[2] | |
| | N (%) of PMDs | N (%) of PMDs | OR (95% CI) | p | OR (95% CI) | p |
| Overall | 277 (0.7) | 536 (2.8) | 3.51 (2.95,4.18) | <0.001 | 3.68 (3.07,4.41) | <0.001 |
| *By time of diagnosis* | | | | | | |
| Prenatal depression | 201 (0.8) | 373 (2.9) | 3.57 (2.89,4.41) | <0.001 | 3.79 (3.04,4.73) | <0.001 |
| *By time since pregnancy* | | | | | | |
| First trimester | 133 (0.8) | 249 (3.2) | 3.56 (2.76,4.60) | <0.001 | 3.49 (2.67,4.55) | <0.001 |
| Second-third trimester | 68 (0.7) | 124 (2.5) | 3.57 (2.45,5.21) | <0.001 | 4.06 (2.67,6.16) | <0.001 |
| Postnatal depression | 76 (0.7) | 163 (2.5) | 3.52 (2.57,4.82) | <0.001 | 3.68 (2.66,5.09) | <0.001 |
| *By time since delivery* | | | | | | |
| ≤6 months | 31 (0.5) | 78 (2.5) | 4.35 (2.63,7.18) | <0.001 | 4.46 (2.65,7.51) | <0.001 |
| 7–12 months | 45 (0.7) | 85 (2.5) | 3.43 (2.22,5.30) | <0.001 | 3.70 (2.33,5.89) | <0.001 |
| **PND and subsequent risk of PMDs** | | | | | | |
| | Women without PND | Women with PND | Model 1[1] | | Model 3[3] | |
| | N (IR) of PMDs | N (IR) of PMDs | HR (95% CI) | p | HR (95% CI) | p |
| Overall | 1,375 (5.2) | 952 (7.9) | 1.60 (1.40,1.82) | <0.001 | 1.58 (1.38,1.80) | <0.001 |
| *By time of diagnosis* | | | | | | |
| Prenatal depression | 935 (5.2) | 590 (7.4) | 1.44 (1.23,1.69) | <0.001 | 1.41 (1.16,1.73) | <0.001 |
| *By time since pregnancy* | | | | | | |
| First trimester | 524 (5.1) | 309 (7.6) | 1.32 (1.05,1.66) | 0.018 | 1.19 (0.88,1.62) | 0.257 |
| Second-third trimester | 358 (4.9) | 281 (7.3) | 1.59 (1.26,1.99) | <0.001 | 1.64 (1.25,2.16) | <0.001 |
| Postnatal depression | 440 (5.1) | 362 (8.8) | 1.88 (1.51,2.36) | <0.001 | 1.93 (1.47,2.54) | <0.001 |
| *By time since delivery* | | | | | | |
| ≤6 months | 207 (5.1) | 157 (8.3) | 1.78 (1.27,2.50) | <0.001 | 1.65 (1.10,2.49) | <0.001 |
| 7–12 months | 217 (5.0) | 205 (9.2) | 2.04 (1.51,2.77) | <0.001 | 2.40 (1.60,3.61) | <0.001 |

CIs, confidence intervals; HR, hazard ratio; IR, incidence rate, per 1000 person-years; N, number; OR, odds ratio; PMDs, premenstrual disorders; PND, perinatal depression.

Analyses were stratified on full sister sets. ORs/HRs and *p*-values were obtained from conditional logistic and Cox regressions.

[1]Model 1 was adjusted for maternal age and calendar year of the delivery.

[2]Model 2 was additionally adjusted for country of birth (Sweden or not), educational level (primary, high school, college and beyond), region of residence (south, middle, or north of Sweden), and cohabitation status (yes or no) at matching.

[3]Model 3 was additionally adjusted for parity (1, and ≥2), BMI during early pregnancy (categorized into <18.5, 18.5 to 24.9, 25 to 29.9, and ≥30 kg/m$^2$), and smoking before pregnancy (no smoking, 1–9, and ≥10 cigarettes per day) and history of psychiatric disorders before pregnancy (yes or no).

known that primiparous women have a higher risk of PND [41] while parity is positively associated with PMDs [42]. It is plausible that premenstrual symptoms can worsen after pregnancy likely due to an escalated abnormal response to hormonal changes in relation to pregnancy [15]. However, it is unclear whether multiple pregnancies would further deteriorate the pathological response to hormone fluctuations. Future studies are needed to better understand the potential mechanisms.

Our work illustrates a higher risk of PMDs among women who experienced PND. Although many women with PMDs have symptom onset in adolescence [43], symptom worsening has been reported with increasing age [44] and parity [15]. It is possible that women with milder premenstrual symptoms experienced worse symptoms after pregnancy and are therefore first diagnosed with PMD after pregnancy. The delayed diagnosis of PMD could be one reason for this finding [45]. Interestingly, we noted a stronger association between PMDs

and subsequent PND than the association in the other direction. This might be because many PMDs have an early onset [43], likely before the average age at first childbirth, whereas we targeted the late-onset PMDs in the cohort study. It is also plausible that women with PND are more likely to take antidepressants, which may mitigate premenstrual symptoms. On the other direction, although women with PMDs may use antidepressants as well, women are more likely to discontinue psychotic medications during pregnancy and even after due to breastfeeding [46,47]. However, future studies are warranted to disentangle the role of treatment in the differential associations observed for both directions.

There are several explanations to the bidirectional association between PMDs and PND. First, both PMDs and PND have shared liability with psychiatric disorders [35,36], which may explain the findings. PMDs could also lead to the development of psychiatric comorbidities [48], which increase the risk of PND. In this scenario, psychiatric disorders are mediators of the studied associations. Indeed, additional adjustment of psychiatric disorders attenuated the associations. However, the bidirectional associations (relative risks) remained robust and even stronger among women without a history of psychiatric disorders, suggesting our findings cannot be entirely explained by psychiatric disorders. On the other hand, although the absolute risk (e.g., probability or incidence rate) of PMDs is higher among women with a psychiatric history, a diagnosis of PND does not translate into an increased risk of PMDs to the same extent as it does for those without a psychiatric history. This is likely because the already heightened risk of PMDs among women with a psychiatric history, particularly among those with a history of depression. However, PMDs or PND have a relatively weaker correlation with other psychiatric disorders compared to depression [35,36]. Among women with a history of other psychiatric disorders, PMDs appeared to confer a higher risk of PND (Table 3) and vice versa (Table 5). Second, PMDs and PND may share other risk factors such as obesity and smoking [4,49,50]. However, the bidirectional association persisted after adjustment of BMI and smoking. Childhood adversities could be another shared risk factor [51,52]. However, sibling comparison should to some extent have addressed that, at least with respect to adversities shared between full sisters. Third, PMDs and PND may share genetic susceptibility. The twin heritability was estimated to be around 54% for PND [53] and 44% to 95% for PMDs [54–56]. Indeed, the attenuation of the associations in sibling comparison lends support to the shared genetic factors and/or familial environmental factors between both disorders. But the associations remained despite the attenuation in sibling comparison. Last, PMDs and PND may share common etiology. The symptom onset of both disorders is linked to hormonal fluctuations, particularly of estrogen and progesterone, which have receptors in the brain, and have been linked to mood alterations [57,58]. PND occurs during a period that is marked with rapid increase of steroid hormones during pregnancy and a rapid decline after delivery [7]. Similarly, the onset of PMD symptoms typically follows the rapid withdraw of hormones in the late luteal phase [8]. It is plausible that an abnormal response to natural hormone fluctuations predisposes women to both PMDs and PND. Future research is, however, needed to reveal the potentially shared underlying etiology of both conditions.

The strength of the study lies in the large sample size with long and complete follow-up, the prospectively and independently collected data on PMDs and PND, and covariates that could confound the studied association. The nested-case control study in combination with the transition into a matched cohort study with onward follow-up allowed us to study the association between PMDs and PND in a bidirectional fashion with efficiency, which is equivalent to two independent cohort studies within the same study population. Moreover, the sibling comparisons allowed us to address the influence of familial factors. However, the study has several limitations. First, the clinical diagnosis of PMDs in the Swedish Patient Register has not been validated. Although prospective daily symptom ratings for at least two consecutive menstrual

cycles are required for diagnosing PMDs in Sweden according to the guidelines [33], we cannot confirm that the clinical decision to assign a diagnosis or medical treatment to every single ascertained PMD in the registers is based on prospective daily ratings. However, clinical guidelines are often well followed owing to the state-funded nature of the public healthcare system in Sweden. Moreover, the Swedish Patient Register has fairly high validity in general [59], with the overall positive predicted value for most diagnoses ranging from 85% to 95% [60]. For a range of psychiatric disorders [61–64] and gynecologic diseases [22,65], the diagnosis has been reported to have high validity. Lastly, sensitivity analysis restricting to PMDs with at least two specialist-made diagnoses at least 28 days apart yielded similar results. In addition, the diagnostic criteria for PMDs may have changed over time. However, stratification analysis by calendar year showed similar results, suggesting that our results are robust given the changes of labeling and diagnostic criteria for PMDs over time. Second, using prescription of antidepressants to identify PND cases might result in misclassifications because antidepressants are also prescribed for other psychiatric disorders. However, sensitivity analysis restricted to clinically diagnosed PND cases showed comparable results. Moreover, reverse causation may to some extent contribute to the observed associations, although we tried to minimize the risk for such bias through the study design that simulates a prospective approach. For instance, some individuals might already have PMDs before pregnancy but received the diagnosis when seeking healthcare for PND. However, similar results have been found when starting the follow-up 1 year after delivery, when such prevalent PMDs would presumably have been captured sooner after the delivery during the postpartum checkups. Third, we did not have data on the exact date of menstruation return for postpartum women, which could be individually different and affected by multiple factors including breastfeeding practices and mode of delivery. Nevertheless, sensitivity analysis with different starting points of the follow-up yielded similar results. Lastly, relying on the Patient Register, we would have missed cases with less severe symptomology and did not seek healthcare service. Moreover, with the ICD code we used to identify PND, we might have missed a small number of mood disturbances or affective disorders that are not sufficiently severe or long-lasting to be classified as depressive episodes. Our findings thus may not generalize well to women with mild PMDs or PND symptomology.

In conclusion, our findings shed light on the bidirectional association between PMDs and PND, supporting a shared underlying etiology. Preconception and maternity care providers should be aware of the risk of developing PND among women with a history of PMDs. Moreover, healthcare providers may inform women with PND about the potential risk of PMDs when menstruation returns after childbirth. The bidirectional relationship is, to a limited extent, explained by psychiatric comorbidities and familial confounding.

## Supporting information

**S1 Checklist. STROBE Statement—Checklist of items that should be included in reports of observational studies.**
(DOCX)

**S1 Methods. Data origin and rational of the choice of covariates.**
(DOCX)

**S1 Fig. Flow chart.**
(DOCX)

**S1 Table. International Classification of Diseases codes used to define the studied medical conditions.**
(DOCX)

**S2 Table. Association of premenstrual disorders (PMDs) with subsequent risk of perinatal depression (PND) adjusted for mediators: A nested case-control study.**
(DOCX)

**S3 Table. Stratified association of premenstrual disorders (PMDs) with subsequent perinatal depression (PND): A nested case control study.**
(DOCX)

**S4 Table. Stratified association of perinatal depression (PND) with subsequent premenstrual disorders (PMDs): A matched cohort study.**
(DOCX)

**S5 Table. Bidirectional link between perinatal depression (PND) with premenstrual disorders (PMDs): Sensitivity analysis.**
(DOCX)

**S6 Table. Association of perinatal depression (PND) with subsequent premenstrual disorders (PMDs): A matched cohort study with different start of follow-up.**
(DOCX)

## Author Contributions

**Conceptualization:** Qian Yang, Unnur A. Valdimarsdóttir, Donghao Lu.

**Data curation:** Qian Yang, Anna Sara Oberg, Donghao Lu.

**Formal analysis:** Qian Yang.

**Funding acquisition:** Qian Yang, Unnur A. Valdimarsdóttir, Donghao Lu.

**Investigation:** Qian Yang.

**Methodology:** Qian Yang, Arvid Sjölander, Anna Sara Oberg, Unnur A. Valdimarsdóttir, Donghao Lu.

**Project administration:** Qian Yang, Unnur A. Valdimarsdóttir, Donghao Lu.

**Resources:** Qian Yang, Fang Fang, Anna Sara Oberg, Unnur A. Valdimarsdóttir, Donghao Lu.

**Software:** Qian Yang, Donghao Lu.

**Supervision:** Arvid Sjölander, Fang Fang, Unnur A. Valdimarsdóttir, Donghao Lu.

**Validation:** Qian Yang, Arvid Sjölander, Donghao Lu.

**Visualization:** Qian Yang.

**Writing – original draft:** Qian Yang.

**Writing – review & editing:** Qian Yang, Emma Bränn, Elizabeth R. Bertone- Johnson, Arvid Sjölander, Fang Fang, Anna Sara Oberg, Unnur A. Valdimarsdóttir, Donghao Lu.

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
