## [Editor Report · Decision Letter 0]

23 May 2023

Dear Dr Yang, 

Thank you for submitting your manuscript entitled "The bidirectional association between premenstrual disorders and perinatal depression: a nationwide register-based study" for consideration by PLOS Medicine.

Your manuscript has now been evaluated by the PLOS Medicine editorial staff and I am writing to let you know that we would like to send your submission out for external peer review.

Please re-submit your manuscript within two working days, i.e. by May 25 2023 11:59PM.

Kind regards,

Philippa Dodd, MBBS MRCP PhD

PLOS Medicine

---

## [Decision Letter · Decision Letter 1]

26 Oct 2023

Dear Dr. Yang,

Thank you very much for submitting your manuscript "The bidirectional association between premenstrual disorders and perinatal depression: a nationwide register-based study" (PMEDICINE-D-23-01426R1) for consideration at PLOS Medicine. 

[LINK]

In light of these reviews, I am pleased to let you know that we would like to consider a revised version that addresses the reviewers' and editors' comments. We won't be able to make a decision about publication until we have seen the revised manuscript and your response, and we plan to seek re-review by one or more of the reviewers. 

We expect to receive your revised manuscript by Nov 16 2023 11:59PM. Please email us (plosmedicine@plos.org) if you have any questions or concerns.

We look forward to receiving your revised manuscript. If you have any questions please don't hesitate to contact me directly via the email address detailed below.

Best wishes,

Pippa

Philippa Dodd, MBBS MRCP PhD

PLOS Medicine

plosmedicine.org

pdodd@plos.org

COMMENTS FROM THE EDITORS

GENERAL

Please respond to all editor and reviewer comments detailed below in full.

Please ensure that the study is reported according to the STROBE guideline, and include the completed STROBE checklist as Supporting Information. Please add the following statement, or similar, to the Methods: "This study is reported as per the Strengthening the Reporting of Observational Studies in Epidemiology (STROBE) guideline (S1 Checklist)."

When completing the checklist, please use section and paragraph numbers, rather than page numbers as these often change in the event of publication.

* The editorial team agree with the reviewer (please see below) regarding the need to better justify the rationale for the use of both a nested-case control study and a matched cohort study. Please include additional detail as relevant in the introduction, methods, and discussion sections of your manuscript.*

** Please also take to care to ensure that you clearly and accurately differentiate between prospective and retrospective data collection and study design. Please see specific comments under ‘Introduction’ and ‘Discussion’.**

TITLE

Please include reference to Sweden in the title, we suggest, ‘The bidirectional association between premenstrual disorders and perinatal depression: a nationwide register-based study from Sweden’ or similar.

COMPETING INTERESTS

All authors must declare their relevant competing interests per the PLOS policy, which can be seen here:

https://journals.plos.org/plosmedicine/s/competing-interests

For authors with ties to industry, please indicate whether any of the interests has a financial stake in the results of the current study.

ABSTRACT

Please structure your abstract using the PLOS Medicine headings (Background, Methods and Findings, Conclusions).

Please combine the Methods and Findings sections into one section, “Methods and findings”.

Abstract Background: Provide the context of why the study is important, as in the current version. The final sentence should clearly state the study question.

Abstract Methods and Findings:

Please ensure that all numbers presented in the abstract are present and identical to numbers presented in the main manuscript text.

Please (briefly) justify the decision to use both methodological approaches in the study.

Please quantify the main results with 95% CIs and p values. When reporting p values please report as p<0.001 and where higher as p=0.002, for example. Please separate upper and lower CI bounds with commas as opposed to hyphens to prevent confusion with reporting of negative values. Suggest reporting statistical information as follows, ‘(OR 4.58; 95% CI [4.28,4.90];p</=) to improve accessibility and clarity for the reader.

Please define ‘OR’, ‘IR’ and ‘HR’ at first use for the reader.

Please include any important dependent variables that are adjusted for in the analyses.

Please include the actual amounts and/or absolute risk(s) of relevant outcomes, not just relative risks or correlation coefficients. (example for absolute risks: PMID: 28399126). 

In the last sentence of the Abstract Methods and Findings section, please describe the main limitation(s) of the study's methodology.

Abstract Conclusions:

Please address the study implications without overreaching what can be concluded from the data; the phrase "In this study, we observed ..." may be useful.

Please interpret the study based on the results presented in the abstract, emphasizing what is new without overstating your conclusions.

Please avoid vague statements such as "these results have major implications for policy/clinical care". Mention only specific implications substantiated by the results.

Please avoid assertions of primacy ("We report for the first time....")

AUTHOR SUMMARY

At this stage, we ask that you include a short, non-technical Author Summary of your research to make findings accessible to a wide audience that includes both scientists and non-scientists. The authors summary should consist of 2-3 succinct bullet points under each of the following headings:

• Why Was This Study Done? Authors should reflect on what was known about the topic before the research was published and why the research was needed.

• What Did the Researchers Do and Find? Authors should briefly describe the study design that was used and the study’s major findings. Do include the headline numbers from the study, such as the sample size and key findings. 

• What Do These Findings Mean? Authors should reflect on the new knowledge generated by the research and the implications for practice, research, policy, or public health. Authors should also consider how the interpretation of the study’s findings may be affected by the study limitations. In the final bullet point of ‘What Do These Findings Mean?’, please describe the main limitations of the study in non-technical language.

Author Summary should immediately follow the Abstract in your revised manuscript. This text is subject to editorial change and should be distinct from the scientific abstract. Please see our author guidelines for more information: https://journals.plos.org/plosmedicine/s/revising-your-manuscript#loc-author-summary

INTRODUCTION

In the discussion you make the distinction between using prospectively collected data as opposed to retrospectively collected data very clear (. Here is it less clear and could be misinterpreted. At first read I thought you were describing your study as prospective. Please revise for clarity

Perhaps instead you could introduce the rationale for using a nested-case control and matched cohort design to address your questions in context of other existing study designs. What advantage does this offer above existing published studies? Please also see the methods section and reviewer comments (below) in reference to the same. You may decide that all this information is better placed there. We leave it to your discretion.

In revising your introduction, please ensure that you indicate whether your study is novel and how you determined that and as in the current version, please conclude the Introduction with a clear description of the study question or hypothesis.

METHODS and RESULTS

As above please ensure that you add the following statement, or similar, to the Methods: "This study is reported as per the Strengthening the Reporting of Observational Studies in Epidemiology (STROBE) guideline (S1 Checklist)."

As above, please clearly explain the rationale for implementing a nested-case control and a matched cohort design.

Line 81 – suggest, ‘Databases’ perhaps, instead, as a sub-heading.

Please see reviewer comments below regarding the use of DSM-5 and ICD-10 coding and the potential implications this has on the construct of your data set thus the outcomes.

Line 117 – what is ‘PDR’? Please define, apologies if I have missed it previously.

As for the abstract, please quantify the main results with 95% CIs and p values. When reporting p values please report as p<0.001 and where higher as p=0.002, for example. When a p value is given, please specify the statistical test used to determine it.

Please separate upper and lower CI bounds with commas as opposed to hyphens to prevent confusion with reporting of negative values. Suggest reporting statistical information as follows, ‘(OR 4.58; 95% CI [4.28,4.90];p</=) to improve accessibility and clarity for the reader.

Please define the length of follow up (eg, in mean, SD, and range).

TABLES

Please provide a table showing the baseline characteristics of the study population in the main manuscript, this is currently placed in the supporting information.

Thank you for indicating that your analyses are adjusted and the factors which are adjusted for. To help facilitate transparent data reporting please also include unadjusted analyses for comparison.

Throughout, as for the main manuscript. Where 95% CIs are reported please use commas as opposed to hyphens to separate upper and lower bounds. 

Throughout, as for the main manuscript, where 95% CIs are reported please also report p values as <0.001 and where higher the exact p value as p= 0.002, for example.

Where p values are reported in the footnote(s) please detail the statistical test used to determine them.

DISCUSSION

Please temper the language in your opening paragraph, ‘…in this study we found…’ might be helpful.

Please present and organize the Discussion as follows: a short, clear summary of the article's findings; what the study adds to existing research and where and why the results may differ from previous research; strengths and limitations of the study; implications and next steps for research, clinical practice, and/or public policy; one-paragraph conclusion.

Line 260 – ‘260 women without psychiatric history of .’ this sentence is incomplete, please revise

Line 265 – ‘..the study is the first…’ claims of primacy can be risky, suggest, ‘to our knowledge’ or similar.

As above (for the introduction) please take care when describing/discussing/differentiating retrospective Vs prospective data collection and study design. Importantly, prospectively collected data often contributes to retrospective study designs.

Line 270 – please use a comma instead of a hyphen to separate upper and lower CI bounds. Please also indicate to the reader that the numerical values represent CIs. Please use formatting as detailed above under abstract and methods/results. 

Line 272 – perhaps ‘more than three times the risk’ instead, to improve accessibility for the reader.

Your discussion of the study strengths could be more detailed. As above, in reference to the dual study design used to answer your questions what advantages does this offer? Please discuss.

Line 316 – please remove the subheading ‘Limitations’ such that the discussion reads as continuous prose.

Please remove the funding, disclosure and conflict statements form the main manuscript and include only in the manuscript submission form when you re-submit your manuscript. In the event of publication these will be compiled as metadata.

REFERENCES

For in-text reference callouts please place citations in square brackets and preceding punctuation. For example [1,3]. Please note the absence of spaces between citations.

In the bibliography please list up to but no more than 6 author names followed by et al in that event that more than 6 contribute to the study.

Please ensure that web references include an ‘Accessed [date]’

Journal name abbreviations should be those found in the National Center for Biotechnology Information (NCBI) databases. 

Please see our website for other reference guidelines https://journals.plos.org/plosmedicine/s/submission-guidelines#loc-references

SUPPORTING INFORMATION

Please cite your Supporting Information as outlined here: https://journals.plos.org/plosmedicine/s/supporting-information

In the published article, supporting information files are accessed only through a hyperlink attached to the captions. For this reason, you must list captions at the end of your manuscript file. You may include a caption within the supporting information file itself, as long as that caption is also provided in the manuscript file. Do not submit a separate caption file.

Please ensure that all abbreviations are defined for the reader at first use.

Please ensure that tables follow out guidance outlined above as for the main manuscript, including the provision of p values and unadjusted analyses, and use of commas to separate upper and lower CI bounds.

As above, please include the completed STROBE checklist as Supporting Information. When completing the checklist, please use section and paragraph numbers, rather than page numbers as these often change in the event of publication. Please be reminded to add the following statement, or similar, to the Methods: "This study is reported as per the Strengthening the Reporting of Observational Studies in Epidemiology (STROBE) guideline (S1 Checklist)."

Please ensure that your supporting information follows the referencing guidance outlined above.

Comments from the reviewers:

Reviewer #1: This is a well-conducted nationwide register-based study on the bidirectional association between premenstrual disorders and perinatal depression. The study design, datasets, statistical methods and analyses, and presentation (tables and figures) and interpretation of the results are mostly adequate and of a good standard. However, still a few issues needing attention.

1) There are 10 matched controls for each index case. Are these controls used for both the nested case-control study and the matched cohort study? or some used for each and at what split?

2) For the Cox models in the matched cohort study, as the outcome is subsequent premenstrual disorders (PMDs) rather than all cause mortality, there is a potential competing risk issue, e.g., from death. Do we have mortality data for the study cohorts? Presumably low, but good to know.

3) Missing data. There is no mention at all on missing data in the study. Are there any? How were the missing data dealt with?

Reviewer #2: This paper is a significant contribution to knowledge of mood disorders affecting menstruating and pregnant people, and the authors should be congratulated for their careful 2-part design to characterize the bidirectional nature of the relationship between menstrually related mood disorders and perinatal depression. A minor comment for your consideration:

-The paper in a couple of places conflates premenstrual disorders (PMD's, which is the basis for your analysis) and the DSM diagnosis of premenstrual dysphoric disorder (PMDD). The criteria you appear to be using for PMDs is more broad than that - the Lancet article you reference (3) is about premenstrual syndrome and refers to findings of prospective and retrospective studies suggest that 5-8% of women with hormonal cycles have moderate to severe symptoms, not the 20-30% you state in this article. In addition, you refer to the Swedish guidelines using DSM-5 criteria requiring prospective rating. However, DSM-5 was first published in 2013, and your study starts in 2001. In your previous paper on PMDs and injury (ref 4) you more accurately characterize that your PMD case ascertainment is not clearly based on prospective rating. You should clarify this distinction in this paper similarly - as written it is misleading to the reader.

[LINK]

---

## [Decision Letter · Decision Letter 2]

18 Dec 2023

Dear Dr. Yang,

Thank you very much for re-submitting your manuscript "The bidirectional association between premenstrual disorders and perinatal depression: a nationwide register-based study from Sweden" (PMEDICINE-D-23-01426R2) for review by PLOS Medicine.

I have discussed the paper with my colleagues and the academic editor and it was also seen again by 3 reviewers. I am pleased to say that provided the remaining editorial and production issues are dealt with we are planning to accept the paper for publication in the journal.

[LINK]

We look forward to receiving the revised manuscript by Dec 21 2023 11:59PM.   

Best wishes,

Pippa

Philippa Dodd, MBBS MRCP PhD

PLOS Medicine

plosmedicine.org

pdodd@plos.org

COMMENTS FROM THE EDITORS:

GENERAL

Thank you for your detailed and considered responses to previous editor and reviewer comments. Please see below for further comments that we require you address prior to publication.

AUTHOR SUMMARY

Thank you for including an author summary which reads very nicely but is too long. Each sub-heading should precede 2-3 concise (1-2 sentence) bullet point statements (much of the statistical information could be removed to conserve space, for example). Please keep in mind that the summary should be accessible to a wide audience including the lay person. Please revise for brevity, in mind of the below guidance:

The author summary should consist of 2-3 succinct bullet points under each of the following headings:

• Why Was This Study Done? Authors should reflect on what was known about the topic before the research was published and why the research was needed.

• What Did the Researchers Do and Find? Authors should briefly describe the study design that was used and the study’s major findings. Do include the headline numbers from the study, such as the sample size and key findings. 

• What Do These Findings Mean? Authors should reflect on the new knowledge generated by the research and the implications for practice, research, policy, or public health. Authors should also consider how the interpretation of the study’s findings may be affected by the study limitations. In the final bullet point of ‘What Do These Findings Mean?’, please describe the main limitations of the study in non-technical language.

This text should be distinct from the scientific abstract. Please see our author guidelines for more information: https://journals.plos.org/plosmedicine/s/revising-your-manuscript#loc-author-summary and please see our website for published examples https://journals.plos.org/plosmedicine/

ABSTRACT

Line 32 – We agreed that case-control design being “equivalent to a cohort study but with improved efficiency” is not an accurate description. Please remove this statement. 

We suggest, ‘…transitioned that design into a matched cohort study with onward follow-up to simulate a prospective study design and examine the risk of PMDs after PND…’. Please also see below for the same/similar elsewhere and amend accordingly and consistently throughout as necessary.

METHODS and RESULTS

Line 196 – ‘…to prospectively examine the risk…’ we understand what you mean but it is technically incorrect as all data is examined (in your study) in retrospect. Please revise suggest (as above), ‘to examine the risk…in a manner which simulates a prospective approach’

TABLES

Table 1 – it would be worth indicating whether there are significant differences between the PND and control groups? Please include p values reporting a p<0.001 and where higher the exact p value as 0.002, for example.

Tables 2-6 – please separate upper and lower CI bounds with commas as opposed to hyphens as the latter can be confused with reporting of negative values. Please include the statistical test used to determine p values in the footnotes.

DISCUSSION

Thank you for your consideration of not overstating your findings. We do think that wider discussion of the implications of your study (in context of the limitations you highlight) and next steps for research would be helpful. How does this study specifically inform subsequent? 

Lines 428 onwards are repeated at lines 430 onwards please correct the duplication. 

Line 436 – ‘prospective study design’ as above please revise and please check throughout for consistency, clarity and accuracy of reporting.

REFERENCES

Please check all references for accuracy as per reviewer comments which we agree with – ref #33 appears incomplete, please revise. Refs# 40 and 48 are detailed as ‘invalid citations’. This list is not exhaustive, please check carefully and amend in accordingly.

Ref #25 – is listed as ‘in press’ and appears not to have been published. Papers cannot be listed in the reference list until they have been accepted for publication or are publicly available on a preprint archive. Please clarify whether the paper has been accepted for publication in the BMJ, as your bibliography might suggest, and please provide a copy for the editors for reference as well as a letter confirming intent to publish. If you are unable to do the former the information may be cited in the text as a personal communication with the author if the author provides written permission to be named. Alternatively, please provide a different appropriate reference.

Please update your reference formatting in-line with our guidance which can be found here https://journals.plos.org/plosmedicine/s/submission-guidelines#loc-references

In the bibliography, please ensure that you list up to but no more than 6 author names followed by et al.

For all web references please ensure you include an, ‘Accessed [date].’

Journal name abbreviations should be those listed in the National Center for Biotechnology Information (NCBI) databases.

SUPPORTING INFORMATION

References - Please ensure that all reference formatting is applied to the supporting information as for the main manuscript. For in-text reference callouts please place citations in square parentheses separate by commas. For example, [1,3,6] or [1-3]. Please check and amend throughout the supporting files.

STROBE Checklist – thank you for including the checklist please amend to refer to section and paragraph numbers as opposed to page (or line numbers) as these often change at publication. Please also amend the column header to read ‘Section/paragraph’ or similar.

Tables – as for the main manuscript please separate upper and lower CI bounds with commas as opposed to hyphens as the latter can be confused with reporting of negative values. Please include the statistical test used to determine p values in the footnotes.

SOCIAL MEDIA

To help us extend the reach of your research, please detail any X (formerly Twitter) handles you wish to be included when we tweet this paper (including your own, your coauthors’, your institution, funder, or lab) in the manuscript submission form when you re-submit the manuscript.

COMMENTS FROM THE REVIEWERS:

Reviewer #1: Thanks authors for their great effort to improve the manuscript. I am satisfied with the response and revision. No further issues needing attention. 

Reviewer #3: The authors have taken a substantial revision of the manuscript which I believe it now meets the standard of PLoS Medicine for publication. Please address the following minor comments before publication.

1. There are 3 references that are not shown correctly in the bibliography list. Please doublecheck to correct it.

2. regarding PNDPMDs - Line 241 of the Revision 1 version of manuscript, Table 4: how to explain the positive results among those WITH history of other psychiatric disorders (new finding no. 4)? Please elaborate on this.

[LINK]

---

## [Editor Report · Decision Letter 3]

19 Feb 2024

Dear Dr Yang, 

On behalf of my colleagues and the Academic Editor, Professor Mark Tomlinson, I am pleased to inform you that we have agreed to publish your manuscript "The bidirectional association between premenstrual disorders and perinatal depression: a nationwide register-based study from Sweden" (PMEDICINE-D-23-01426R3) in PLOS Medicine.

Prior to publication we require that you address the following:

Line 68 – please revise this point into 2 bullet points as detailed below:

* We conducted a nested case-control study and transitioned that design into a matched cohort study with onward follow-up to simulate a prospective study design. 

* Among approximately 1.8 million singleton pregnancies in Sweden between 2001-2018, we identified 84,949 women with PND and 849,482 unaffected women, individually matched on age and calendar year. Pregnancies from full sisters of women with PND were also identified for sibling comparison.

Line 76 – please remove the word ‘their’

Line 83 – please include a final bullet point in the ‘what do these findings mean’ sub-section, which describes the main limitations of the study in nontechnical language.

Line 448 – please remove the statement ‘PND women’ and replace with, ‘women with PND’

PRESS

Best wishes,

Pippa 

Philippa Dodd, MBBS MRCP PhD 

PLOS Medicine

pdodd@plos.org